# Deep quantum neural networks on a superconducting processor

Xiaoxuan Pan [1,5], Zhide Lu [1,5], Weiting Wang[1], Ziyue Hua[1], Yifang Xu[1], Weikang Li [1], Weizhou Cai[1], Xuegang Li[1], Haiyan Wang[1], Yi-Pu Song[1], Chang-Ling Zou [2,3], Dong-Ling Deng [1,3,4] ✉ & Luyan Sun [1,3] ✉

Deep learning and quantum computing have achieved dramatic progresses in recent years. The interplay between these two fast-growing fields gives rise to a new research frontier of quantum machine learning. In this work, we report an experimental demonstration of training deep quantum neural networks via the backpropagation algorithm with a six-qubit programmable superconducting processor. We experimentally perform the forward process of the back-propagation algorithm and classically simulate the backward process. In particular, we show that three-layer deep quantum neural networks can be trained efficiently to learn two-qubit quantum channels with a mean fidelity up to 96.0% and the ground state energy of molecular hydrogen with an accuracy up to 93.3% compared to the theoretical value. In addition, six-layer deep quantum neural networks can be trained in a similar fashion to achieve a mean fidelity up to 94.8% for learning single-qubit quantum channels. Our experimental results indicate that the number of coherent qubits required to maintain does not scale with the depth of the deep quantum neural network, thus providing a valuable guide for quantum machine learning applications with both near-term and future quantum devices.

Machine learning has achieved tremendous success in both commercial applications and scientific research over the past decade. In particular, deep neural networks play a vital role in cracking some notoriously challenging problems, ranging from playing Go[1] to predicting protein structures[2]. They contain multiple hidden layers and are believed to be more powerful in extracting high-level features from data than traditional methods[3,4]. The learning process can be fueled by updating the parameters through gradient descent, where the backpropagation (BP) algorithm enables efficient calculations of gradients via the chain rule[3].

By harnessing the weirdness of quantum mechanics such as superposition and entanglement, quantum machine learning approaches hold the potential to bring advantages compared with their classical counterpart. In recent years, exciting progress has been made

along this interdisciplinary direction[5–10]. For example, rigorous quantum speedups have been proved in classification models[11] and generative models[12] with complexity-theoretic guarantees. In terms of the expressive power of quantum neural networks, there is also preliminary evidence showing their advantages over the comparable feedforward neural networks[13]. Meanwhile, noteworthy progress has also been made on the experimental side[14–22]. For example, in ref. 14, the authors realize a quantum convolutional neural network on a superconducting quantum processor. In ref. 15, an experimental demonstration of quantum adversarial learning has been reported. Similar to deep classical neural networks with multiple hidden layers, a deep quantum neural network (DQNN) with the layer-by-layer architecture is proposed[23–25], which can be trained via a quantum analog of the BP algorithm. The word "deep" in the DQNN refers to multiple

[1]Center for Quantum Information, Institute for Interdisciplinary Information Sciences, Tsinghua University, Beijing 100084, China. [2]CAS Key Laboratory of Quantum Information, University of Science and Technology of China, Hefei, Anhui 230026, China. [3]Hefei National Laboratory, Hefei 230088, China. [4]Shanghai Qi Zhi Institute, No. 701 Yunjin Road, Xuhui District, Shanghai 200232, China. [5]These authors contributed equally: Xiaoxuan Pan and Zhide Lu. ✉e-mail: dldeng@tsinghua.edu.cn; luyansun@tsinghua.edu.cn

hidden layers, rather than the large depth of quantum circuits. Under this framework, the quantum analog of a perceptron is a general unitary operator acting on qubits from adjacent layers, whose parameters are updated by multiplying the corresponding updating matrix of the perceptron in the training process.

In this paper, we report an experimental demonstration of training DQNNs through the BP algorithm on a programmable superconducting processor with six frequency-tunable transmon qubits. We find that a three-layer DQNN can be efficiently trained to learn a two-qubit target quantum channel with a mean fidelity of up to 96.0% and the ground state energy of molecular hydrogen with an accuracy of up to 93.3% compared to the theoretical prediction. In addition, we also demonstrate that a six-layer DQNN can efficiently learn a one-qubit target quantum channel with a mean fidelity of up to 94.8%. Our approach can carry over to other DQNNs with a larger width and depth straightforwardly, thus paving the way towards large-scale quantum machine learning with potential advantages in practical applications.

## Results

### Deep quantum neural networks

As sketched in Fig. 1a, our DQNN has a layer-by-layer structure, and maps the quantum information layerwise from the input layer state $\rho^{in}$, through $L$ hidden layers, to the output layer state $\rho^{out}$. Quantum perceptrons are the building blocks of the DQNN, and different types of quantum perceptrons have been experimentally implemented recently[26–28]. DQNNs with the most general form of quantum perceptrons, which can apply generic unitaries on all qubits at adjacent layers, are capable of universal quantum computation[29]. In practice, we usually employ restricted forms of perceptrons for experimental implementations on noisy quantum devices. In this work, a single quantum perceptron is defined as a parameterized quantum circuit applied to the corresponding qubit pair at adjacent layers, which is

shown in Fig. 1b. A sequential combination of the quantum perceptrons constitutes the layerwise operation between adjacent layers. One of the key characteristics of the DQNN is the layer-by-layer quantum state mapping, allowing efficient training via the quantum BP algorithm[23].

We sketch the general experimental training process in Fig. 1c. When performing the quantum BP algorithm, one only requires the information from adjacent two layers, rather than the full DQNN, to evaluate the gradients with respect to all parameters at these two layers. Such a BP-equipped DQNN bears the following merit: the number of coherent qubits required to maintain does not scale with the depth of the DQNN. This merit makes it possible to realize DQNNs with reduced number of layers of qubits through qubit reusing[23]. The qubits can be reused as follows: after completing the operations between adjacent layers, we can reset qubits in the previous layer to the fiducial product state, and then reuse them as qubits in the subsequent layer. We note that resetting qubits takes extra time in experiments, and this raises additional coherence requirements for qubits in the current layer. By reusing qubits, only two layers of qubits are required to implement a DQNN regardless of its depth. The experimental errors on qubits in the previous layer will affect qubits in the current layer, which will limit the depth of the DQNN in real experiments with noisy devices.

### Experimental setup

Our experiment is carried out on a superconducting quantum processor, which possesses six two-junction and frequency-tunable transmon qubits[30–37]. As photographed in Fig. 1d, the chip is fabricated with the layout of the qubits being purposely and carefully optimized for a layer-by-layer structure. Each transmon qubit is coupled to an individual flux control line, XY control line, and quarter-wavelength readout resonator, respectively. All readout resonators are

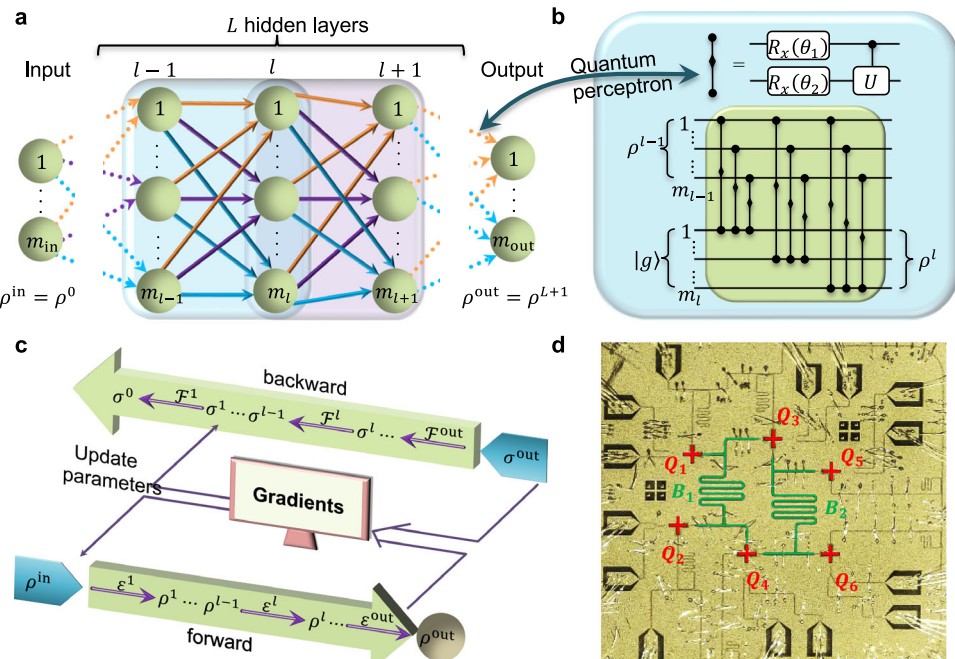

**Fig. 1 | A schematic of training deep quantum neural networks. a** Architecture exhibition of a general DQNN. Information propagates layerwise from the input layer to the output layer. At adjacent two layers, we apply the quantum perceptron in the order according to the exhibited circuit in **b**. A quantum perceptron is realized by applying two single-qubit rotation gates $R_x(\theta_1)$ and $R_x(\theta_2)$ (the rotations along the $x$ axis with variational angles $\theta_1$ and $\theta_2$, respectively) followed by a fixed two-qubit controlled-Phase gate. **c** Illustration of the quantum backpropagation algorithm. We apply forward channels $\mathcal{E}$ on $\rho^{in}$ and successively obtain $\{\rho^1, \rho^2...\rho^{out}\}$, and apply backward channels $\mathcal{F}$ to successively obtain $\{\sigma^{out}, \sigma^L...\sigma^1\}$ in the backward process. These forward and backward terms are used for the gradient evaluation. **d** Exhibition of a quantum processor with six superconducting transmon qubits, which are used to experimentally implement the DQNNs. The transmon qubits ($Q_1$–$Q_6$) and the bus resonators ($B_1$ and $B_2$) are marked.

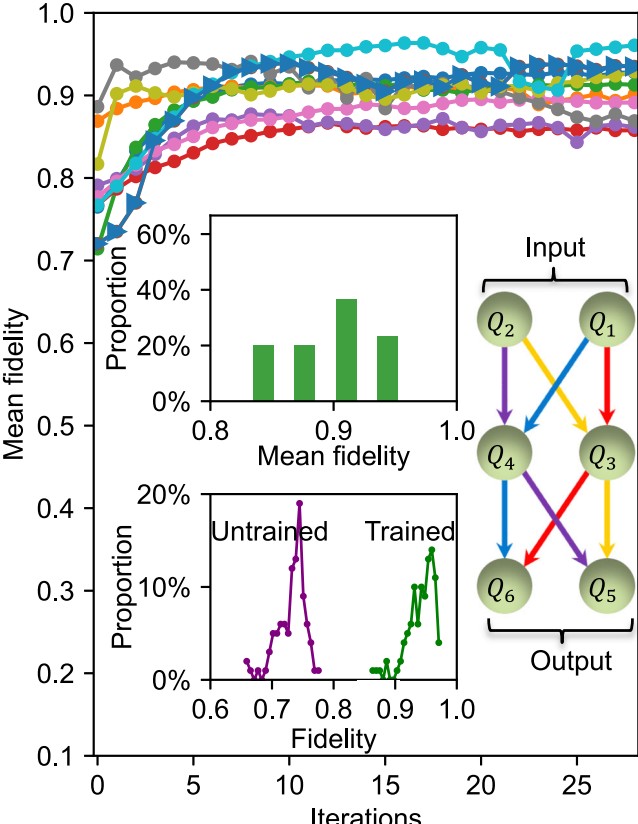

**Fig. 2 | Experimental results for learning a two-qubit quantum channel.** We train the three-layer DQNN$_1$ with 30 different initial parameters and plot the mean fidelity as a function of training iterations for 10 of them for clarity. The upper left inset shows the distribution of the converged mean fidelities of these 30 different initial parameters. We choose one of the learning curves (marked with triangles), then randomly generate 100 different input quantum states, and test the fidelity between their output states given by the target quantum channel and the trained (untrained) DQNN$_1$. In the lower left inset, two curves show the distribution of the fidelities for the trained (untrained) DQNN$_1$. The right inset is a schematic illustration of DQNN$_1$. At adjacent layers, we apply the quantum perceptrons in the order provided in Supplementary Table 2.

coupled to a common transmission line, which is connected through a Josephson parametric amplifier for high-fidelity single-shot readout of the qubits[38,39]. In order to implement the two-qubit gates in the quantum perceptrons, two separate half-wavelength bus resonators are respectively used to mediate the interactions among the qubits between layers[16,40,41]. The characterized fidelities of single-qubit $R_x$ gates are above 99.5%, while the average fidelity of the two-qubit gates is 98.4%. The detailed experimental setup and device parameters can be found in Supplementary Note 4.

## Learning a two-qubit quantum channel

The first application of DQNNs is learning a quantum channel. Specifically, we consider using DQNNs to learn a two-qubit target quantum channel. We experimentally implement a three-layer DQNN with two qubits in each layer. This three-layer DQNN is denoted by DQNN$_1$. As mentioned above, we employ restricted form of perceptrons due to realistic experimental limitations. In general, the restricted form of perceptrons would reduce the representation power of DQNNs. Therefore, we construct the target quantum channel using the same ansatz as DQNN$_1$ with randomly generated parameters $\theta_t$, so that the target quantum channel would be within the representation range of DQNN$_1$ (see Methods). Here, we choose $|00\rangle$, $|01\rangle$, $|++\rangle$, and $|+i+i\rangle$ as our input states $\rho_x^{in}$, where the subscript $x = 1, 2, 3, 4$ is the labeling, $|0\rangle$

and $|1\rangle$ are the eigenstates of Pauli $Z$ matrix, $|+\rangle$ ($|-\rangle$) is the eigenstate of Pauli $X$ matrix, and $|+i\rangle$ is the eigenstate of Pauli $Y$ matrix. The four pairs of $(\rho_x^{in}, \tau_x^{out})$ serve as the training dataset, where $\tau_x^{out}$ is the corresponding desired output state produced by the target quantum channel. The optimization goal is to maximize the mean fidelity between $\tau_x^{out}$ and the measured DQNN output $\rho_x^{out}$ averaged over all four input states. The general training procedure goes as follows: (1) Initialization: we randomly choose the initial gate parameters $\theta$ for all perceptrons in DQNN$_1$. (2) Forward process (implemented on our quantum processor): for each training sample $(\rho_x^{in}, \tau_x^{out})$, we prepare the input layer to $\rho_x^{in}$, then apply layerwise forward channels $\mathcal{E}^1$ and $\mathcal{E}^{out}$, and extract $\rho_x^1$ and $\rho_x^{out}$ successively by carrying out quantum state tomography[42]. (3) Backward process (implemented on a classical computer): we initialize the output layer to $\sigma_x^{out}$, which is determined by $\rho_x^{out}$ and $\tau_x^{out}$ (see Supplementary Note 1), and then apply backward channels $\mathcal{F}^{out}$ and $\mathcal{F}^1$ on $\sigma_x^{out}$ to successively obtain $\sigma_x^1$ and $\sigma_x^0$. (4) Based on $\{(\rho_x^{l-1}, \sigma_x^l)\}$, we evaluate the gradient of the fidelity with respect to all the variational parameters in the adjacent layers $l-1$ and $l$. Then we take the average over the whole training dataset for the final gradient, which is used to update the variational parameters $\theta$. (5) Repeat (2), (3), (4) for $s_0$ rounds. The pseudocode for our algorithm is provided in Supplementary Note 1.

In Fig. 2, we randomly choose 30 different initial parameters $\theta$, and then train DQNN$_1$ to learn the same target quantum channel. We observe that DQNN$_1$ converges quickly during the training process, with the highest fidelity above 96%. To benchmark the performance of DQNN$_1$, we carry out a classical simulation of the training in Supplementary Note 3, where we train DQNN$_1$ to learn a target quantum channel without considering any experimental imperfections. The numerical results show that the average converged mean fidelity for 50 different initial parameters is above 98%. Compared with the numerical simulation results, the deviation of the final converged fidelities is due to experimental imperfections, including qubit decoherence and residual ZZ interactions between qubits[43–45]. In the upper left inset of Fig. 2, we show the distribution for all the converged fidelities from these 30 repeated experiments. We expect that the distribution will concentrate to a higher fidelity for improved performance of the quantum processor.

To evaluate the performance of DQNN$_1$, we choose one training process from the 30 experiments, and refer the DQNN$_1$ with parameters corresponding to the ending (starting) iteration of the training curve as the trained (untrained) DQNN$_1$. We generate other 100 different input quantum states and experimentally measure their corresponding output states produced by the trained (untrained) DQNN$_1$. We test the fidelity between output states given by the target channel and the trained (untrained) DQNN$_1$. As shown in the lower left inset of Fig. 2, for the trained DQNN$_1$, 43% of the fidelities exceed 0.95 and 95% of the fidelities are higher than 0.9, which separate away from the distribution of the results of the untrained DQNN$_1$. This contrast illustrates the effectiveness of the training process of DQNN$_1$.

## Learning the ground state energy

DQNNs also provide a complementary approach to solving quantum chemistry problems. Here, we apply DQNNs to learn the ground state energy of a given Hamiltonian $H$ as an example. The optimization goal is to minimize the energy estimate $\text{tr}\,(\rho^{out} H)$ for the output state of the DQNN. We aim to learn the ground state energy of the molecular hydrogen Hamiltonian[46]. By exploiting the Bravyi-Kitaev transformation and certain symmetry, the Hamiltonian of molecular hydrogen can be reduced to the effective Hamiltonian acting on two qubits: $\hat{H}_{BK} = g_0 \mathbf{I} + g_1 Z_0 + g_2 Z_1 + g_3 Z_0 Z_1 + g_4 Y_0 Y_1 + g_5 X_0 X_1$, where $X_i, Y_i, Z_i$ are Pauli operators on the $i$-th qubit, and coefficients $g_j$ ($j = 0, \cdots, 5$) depend on the fixed bond length of molecular hydrogen. We consider the bond length 0.075 nm in this work and the corresponding coefficients $g_i$ can be found in ref. 46.

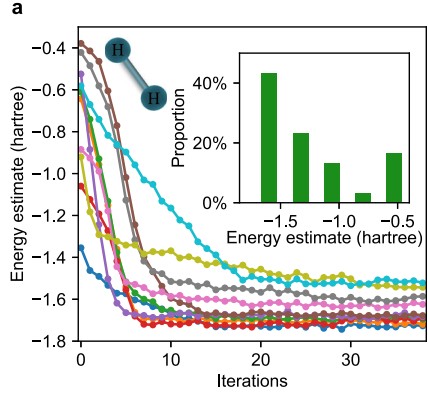

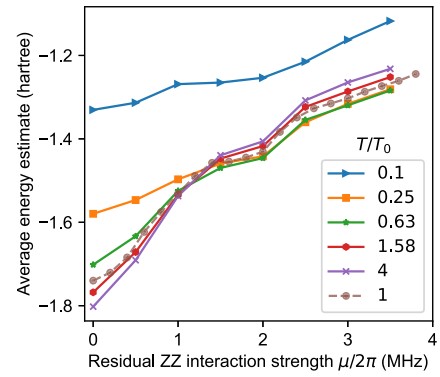

**Fig. 3 | Experimental and numerical results for learning the ground state energy of molecular hydrogen. a** Experimental energy estimate at each iteration during the learning process for different initial parameters. The inset displays the distribution of converged energy estimates of 30 different initial parameters. **b** Numerical results for the mean energy estimate with different coherence time $T$

and residual ZZ interaction strength $\mu$ between qubits. Specifically, we assume the same $\mu$ between all neighboring qubits and adjust the coherence time by the same ratio ($T/T_0$) for all qubits, where $T$ and $T_0$ are the coherence times in the numerical simulation and the experiment, respectively.

We use DQNN$_1$ again as the variational ansatz to learn the ground state of molecular hydrogen with the following procedure, similar to the previous one of learning a quantum channel: (1) Initialization: we prepare the input layer to the fiducial product state $|00\rangle$, and randomly generate initial gate parameters $\boldsymbol{\theta}$ for DQNN$_1$. 2) In the forward process (implemented on the quantum processor), we apply forward channels $\mathcal{E}^1$ and $\mathcal{E}^{out}$ in succession, and extract quantum states of the hidden layer ($\rho^1$) and the output layer ($\rho^{out}$) by quantum state tomography. (3) In the backward process (implemented on a classical computer), we initialize the quantum state of the output layer to $\sigma^{out}$, and then obtain $\sigma^1$ and $\sigma^0$ after successively applying backward channels $\mathcal{F}^{out}$ and $\mathcal{F}^1$ on $\sigma^{out}$. (4) Based on $\{(\rho^{l-1}, \sigma^l)\}$, we calculate the gradient of the energy estimate with respect to all the variational parameters in the adjacent layers $l-1$ and $l$, and then update all gate parameters in DQNN$_1$. (5) Repeat (2), (3), (4) for $s_0$ rounds. The pseudocode for our algorithm is provided in Supplementary Note 1.

We train DQNN$_1$ with 30 different initial parameters and show our experimental results in Fig. 3a. We observe that DQNN$_1$ converges within 20 iterations. The lowest ansatz energy estimate reaches below −1.727 (hartree) in the learning process, with an accuracy up to 93.3% compared to the theoretical value of the ground state energy −1.851 (hartree). This shows the good performance of DQNN$_1$ and the accuracy of our experimental system control. The inset of Fig. 3a shows the distribution of all the converged energy from these 30 repeated experiments with different initial parameters, six of which have accuracy above 90%.

To numerically investigate the effects of experimental imperfections on training DQNNs, we consider two possible sources: decoherence of qubits and residual ZZ interactions between qubits (see Methods for details). Under different residual ZZ interaction strength $\mu$ and different decoherence time $T$, we numerically train the DQNN with 30 different initial parameters to learn the ground state energy of the molecular hydrogen. We find that for four of these initial parameters DQNN$_1$ converges to local minima instead of the global minimum. Excluding these abnormal instances with local minima, we plot the average energy estimate as a function of $\mu$ with different $T/T_0$ in Fig. 3b, where $T_0$ is the experimentally measured qubit coherence time. The numerical results show that, when there is no residual ZZ interaction, the decoherence of the qubits at $T/T_0 = 1$ degrades the accuracy of the average energy estimate by 6% to −1.74 (the leftmost point of the dotted line). At $\mu/2\pi \approx 1$ MHz (close to our experimental characterization) with $T/T_0 = 1$, the average energy estimate is −1.53, which is about 17% higher than the theoretical value and is comparable to the experimental values shown in Fig. 3a. Apparently, such a large $\mu$

dominantly limits the training performance, while the variation of the coherence time has a minor effect. This is anticipated given the fact that the total running time of the DQNN (1.2 μs) is significantly shorter than the average characteristic coherence time of the qubits (7.5 μs). These experimental imperfections can be suppressed after introducing advanced technologies in the design and fabrication of better superconducting quantum circuits, such as tunable couplers[47–49] and tantalum-based qubits[50,51].

## Learning a one-qubit quantum channel

To further illustrate the efficiency of the quantum BP algorithm, we construct another DQNN with four hidden layers (denoted as DQNN$_2$) by rearranging our six-qubit quantum processor into a six-layer structure, with one-qubit respectively in each layer. We focus on the task of learning a one-qubit target quantum channel, which is constructed using the same ansatz as DQNN$_2$ with randomly generated parameters (see Methods). We choose $|0\rangle, |1\rangle, |-\rangle$ as our input states and compare the measured output states of DQNN$_2$ with the desired ones from the target single-qubit quantum channel. The general training procedure is similar as in training DQNN$_1$ discussed above. Our experimental results are summarized in Fig. 4, which shows the learning curves for 10 different initial parameters. We find that DQNN$_2$ can learn the target quantum channel with a mean fidelity up to 94.8%. We notice that the variance among the converged mean fidelity in DQNN$_2$ is smaller than that for DQNN$_1$, which may be attributed to the smaller total circuit depth and thus less error accumulation due to experimental imperfections. To study the learning performance, we choose one of these learning curves (marked in triangles), and refer DQNN$_2$ with parameters corresponding to the ending (starting) iteration of the learning curve as the trained (untrained) DQNN$_2$. We then use other 100 different input quantum states to test the trained and untrained DQNN$_2$ by measuring the fidelities between the experimental output states and the corresponding desired ones given by the target quantum channel. As shown in the upper inset of Fig. 4, the fidelity distribution concentrates around 0.92 for the trained DQNN$_2$, which stands in stark contrast to that of the untrained DQNN$_2$ and thus indicates a good performance after training.

## Discussion

In this work, we experimentally perform the forward process while implementing the backward process on a classical computer. For the task of learning a target quantum channel, we note that the backward process can also be implemented with a quantum device in principle (see Supplementary Note 1 for an experimental proposal). Yet, it is

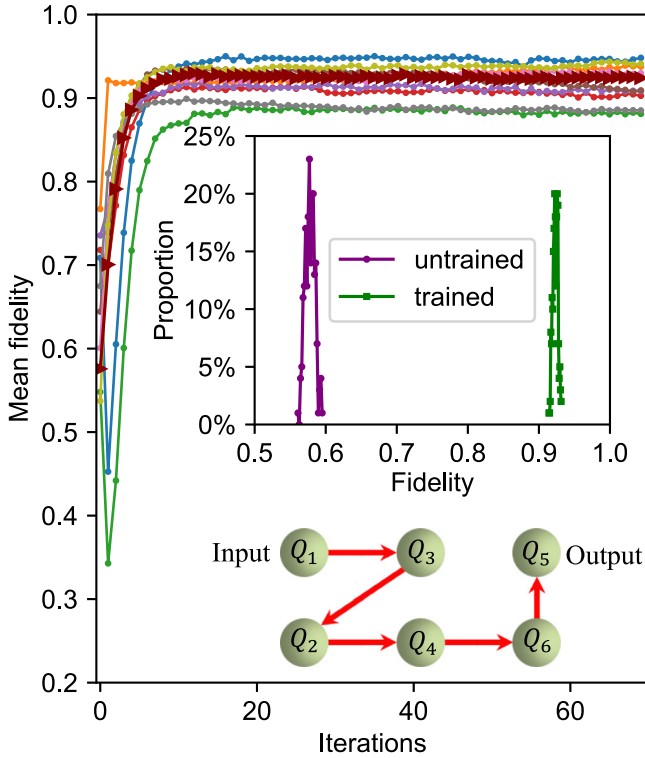

**Fig. 4 | Experimental results for learning a one-qubit quantum channel.** The mean fidelity of training the six-layer DQNN$_2$ is plotted as the function of training iterations for different initial parameters. We randomly generate 100 different single-qubit states, and evaluate the fidelities between their output states produced by DQNN$_2$ and their desired output states given by the target quantum channel. The upper inset displays the distribution in two cases: a well-trained DQNN$_2$ (marked by squares) and an untrained (marked by circles) DQNN$_2$, both are defined with the learning curve marked in triangles. The lower inset is a schematic illustration of DQNN$_2$, where we apply the perceptrons in the order indicated by the direction of the arrows.

more challenging to achieve the required experimental accuracy for backward channels than that for forward channels. In the future, we expect an experimental implementation of the backward channel on a quantum processor with better performance.

The efficiency of the quantum BP algorithm can be measured in terms of the required number of copies for each training data per training iteration (see Supplementary Note 2 for detailed discussions). For the training with the quantum BP algorithm, this number scales exponentially with the number of qubits in the hidden layers and the output layer. This is due to the fact that an exponential number of measurements are required for quantum state tomography and the experimental implementation of backward channels. Compared with the training not utilizing the quantum BP algorithm, we find that the use of the quantum BP algorithm can enhance the training efficiency in some cases where DQNNs have narrow hidden layers. Such DQNNs can be used as quantum autoencoders to compress and denoise quantum data[52].

In summary, we have demonstrated the training of deep quantum neural networks on a six-qubit programmable superconducting quantum processor. We experimentally exhibit its intriguing ability to learn quantum channels and learn the ground state energy of a given Hamiltonian. The quantum BP algorithm demonstrated in our experiments can be directly applied to DQNNs with extended widths and depths. With further improvements in experimental conditions, quantum perceptrons with enhanced expressive power are expected to be constructed with deeper circuits, which allows DQNNs to tackle more challenging tasks in the future.

## Methods

### Framework

We consider a DQNN that includes $L$ hidden layers with a total number $m_l$ of qubits in layer $l$. The qubits in two adjacent layers are connected with quantum perceptrons and each perceptron consists of two single-qubit rotation gates $R_x(\theta_1)$ and $R_x(\theta_2)$ along the $x$ axis with variational angles $\theta_1$ and $\theta_2$, respectively, followed by a fixed two-qubit controlled-Phase gate. The unitary of the quantum perceptron that acts on the $i$-th qubit at layer $l-1$ and the $j$-th qubit at layer $l$ in the DQNN is written as $U^l_{(i,j)}(\theta^l_{(i,j),1}, \theta^l_{(i,j),2})$. Then the unitary product of all quantum perceptrons acting on the qubits in layers $l-1$ and $l$ is denoted as $U^l = \prod_{j=m_l}^{1} \prod_{i=m_{l-1}}^{1} U^l_{(i,j)}$. The DQNN acts on the input state $\rho^{\text{in}}$ and produces the output state $\rho^{\text{out}}$ according to

$$\rho^{\text{out}} \equiv \text{tr}_{\text{in,hid}}\left(\mathcal{U}(\rho^{\text{in}} \otimes |0\cdots0\rangle_{\text{hid,out}}\langle0\cdots0|)\mathcal{U}^\dagger\right), \quad (1)$$

where $\mathcal{U} \equiv U^{\text{out}} U^L U^{L-1} \dots U^1$ is the unitary of the DQNN, and all qubits in the hidden layers and the output layer are initialized to a fiducial product state $|0\cdots0\rangle$. The characteristic of the layer-by-layer architecture enables $\rho^{\text{out}}$ to be expressed as a series of maps on $\rho^{\text{in}}$:

$$\rho^{\text{out}} = \mathcal{E}^{\text{out}}\left(\mathcal{E}^L\left(\dots\mathcal{E}^2\left(\mathcal{E}^1(\rho^{\text{in}})\right)\dots\right)\right), \quad (2)$$

where $\mathcal{E}^l(\rho^{l-1}) \equiv \text{tr}_{l-1}\left(U^l(\rho^{l-1} \otimes |0\cdots0\rangle_l\langle0\cdots0|)U^{l\dagger}\right)$ is the forward quantum channel.

In the Supplementary Information, we prove that for the two machine learning tasks in our work, the derivative of the mean fidelity or the energy estimate with respect to $\theta^l_{(i,j),k}$ can be calculated with the information of layers $l-1$ and $l$, which can be written as $G(\boldsymbol{\theta}^l, \rho^{l-1}, \sigma^l)$ with $\boldsymbol{\theta}^l$ incorporating all parameters in layers $l-1$ and $l$. We note that $\rho^{l-1} = \mathcal{E}^{l-1}(\dots\mathcal{E}^2(\mathcal{E}^1(\rho^{\text{in}}))\dots)$ refers to the quantum state in layer $l-1$ in the forward process, and $\sigma^l = \mathcal{F}^{l+1}(\dots\mathcal{F}^{\text{out}}(\cdots)\dots)$ represents the backward term in layer $l$ with $\mathcal{F}^l$ being the adjoint channel of $\mathcal{E}^l$. The backward channel $\mathcal{F}^l$ applies on the backward term $\sigma^l$ and produces $\sigma^{l-1}$ according to $\sigma^{l-1} = \mathcal{F}^l(\sigma^l) = \text{tr}_l\left(\left(\mathbb{I}_{l-1} \otimes |0\rangle_l\langle0|\right)U^{l\dagger}\left(\mathbb{I}_{l-1} \otimes \sigma^l\right)U^l\right)$.

### Generating random input quantum states

To evaluate the learning performance in the task of learning a target quantum channel, we need to generate many different input quantum states and test the fidelity between their output states produced by DQNN$_1$ and their desired output states given by the target quantum channel.

For the task of learning a two-qubit quantum channel, we generate these input quantum states by separately applying single-qubit rotation gates $R_{a_1}(\Omega_1) \otimes R_{a_2}(\Omega_2)$ on the two qubits initialized in $|00\rangle$. Here each rotation gate has a random rotation axis $a_i$ in the $x-y$ plane and a random rotation angle $\Omega_i$.

For the task of learning a one-qubit quantum channel, we generate the input quantum states by applying single-qubit rotation gates $R_b(\Phi)$ on the input qubit initialized in $|0\rangle$ with a random rotation axis $b$ in the $x-y$ plane and a random rotation angle $\Phi$.

### The target quantum channels

The target quantum channels are constructed using the DQNN ansatz with randomly chosen parameters $\boldsymbol{\theta}_t$. For DQNN$_1$, the schematic illustration is shown in the right inset of Fig. 2. The input state is encoded in qubits $Q_1$ and $Q_2$ (input layer). After completing all the layerwise quantum perceptrons, the output state is obtained from qubits $Q_5$ and $Q_6$ (output layer). Each quantum perceptron is applied with randomly chosen single-qubit rotation angles, which are provided in the third column of Supplementary Table 2. The target quantum channel for DQNN$_2$ is constructed in a similar way, with the corresponding parameters (randomly chosen) also provided in Supplementary Table 2.

## Numerical simulations

We numerically consider two sources for experimental imperfections: decoherence of qubits and residual ZZ interactions between qubits. In our simulation, we consider the effects of the residual ZZ interactions only during the implementation of rotation gates for simplicity. Specifically, $R_{x,i}(\theta)$ (rotation gate $R_x$ on the $i$-th qubit) is realized by the evolution of a time-dependent Hamiltonian of the form $H_0(t) = \theta(a_i^\dagger + a_i)h(t)/2$, where $a_i$ ($a_i^\dagger$) is the corresponding annihilation (creation) operator, and $h(t)$ is the time-dependent driving strength given as a Gaussian function $h(t) = Ae^{-(t/\delta)^2}$. We set the evolution time of the rotation gate as $\Delta t = 40$ ns and $\delta = 10$ ns. The unwanted residual ZZ interactions between qubits are considered by adding a Hamiltonian of the form $H_1(t) = \sum_{\langle i,j\rangle} \mu_{ij} a_i^\dagger a_i a_j^\dagger a_j$ to $H_0(t)$, where $\langle i,j\rangle$ denotes the interaction between nearest-neighbor qubits, and $\mu_{ij}$ is the interaction strength between the $i$-th and $j$-th qubits. The qubit decoherence is considered by adding the relaxation term $\sqrt{1/T_{1,i}}a_i$ and the pure dephasing term $\sqrt{1/T_{\phi,i}}a_i^\dagger a_i$ as the collapse operators to the evolution. Here, $T_{1,i}$ and $T_{\phi,i}$ are the energy relaxation time and the pure dephasing time of the $i$-th qubit, respectively.

## Data availability

The data generated in this study have been deposited in the Figshare database under accession code https://doi.org/10.6084/m9.figshare.22802501[53].

## Code availability

The codes for numerical simulations are available at https://zenodo.org/badge/latestdoi/552193072[54].

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

## Acknowledgements
We thank Wenjie Jiang for the helpful discussions. We acknowledge the support of the National Natural Science Foundation of China (Grants No. 92165209, No. 11925404, No. 11874235, No. 11874342, No. 11922411, No. 12061131011, No. T2225008, No. 12075128), the National Key Research and Development Program of China (Grants No. 2017YFA0304303), Key-Area Research and Development Program of Guangdong Province (Grant No. 2020B0303030001), Anhui Initiative in Quantum Information Technologies (AHY130200), China Postdoctoral Science Foundation (BX2021167), and Grant No. 2019GQG1024 from the Institute for Guo Qiang, Tsinghua University. D.-L.D. also acknowledges additional support from the Shanghai Qi Zhi Institute.

## Author contributions
X.P. carried out the experiments and analyzed the data with the assistance of Z.H. and Y.X.; L.S. directed the experiments; Z.L. formalized the theoretical framework and performed the numerical simulations under the supervision of D.-L.D.; W.L. and C.-L.Z. provided theoretical support; W.C. fabricated the parametric amplifier; W.W. and X.P. designed the devices; X.P. fabricated the devices with the assistance of W.W., H.W., and Y.-P.S.; Z.H., W.C., and X.L. provided further experimental support; X.P., Z.L., W.L., C.-L.Z., D.-L.D., and L.S. wrote the manuscript with feedback from all authors.

## Competing interests
The authors declare no competing interests.
