## [Peer Review File · Nature Communications]

Deep quantum neural networks on a superconducting processorREVIEWER COMMENTS

Reviewer #1 (Remarks to the Author):

The manuscript discusses the implementation and training of quantum neural networks, in which two-qubit gates are implemented between the qubits belonging to different layers. This type of neural network has been introduced in Beer et al Nat Commun. 11, 808 (2020). In fact studies of the trainability and generalization capacities of this type of quantum neural network, <https://arxiv.org/abs/2208.06198>, found that it performs better than other constructions in many aspects.

The present work could thus form a significant step towards the development of quantum neural networks. There are however a number of aspects that need to be clarified before I dare to make a recommendation.

1) The work shows supervised learning for the network to train it to approximate a quantum channel. I however did not find what this quantum channel is. To assess the quality of this achievement it is however important to know what kind of channel this is. In particular, the fidelity of the approximation without training, i.e. for random parameters, is already above 70%. Is the targeted output state a pure state? In that case I would expect the untrained fidelity to be much lower, 25%. Also in the deeper network, the untrained fidelity is above 50%.

2) The authors argue that this structure enhances the capabilities of current hardware since the coherence of the qubits only needs to be maintained as long as the qubits are involved in a two layer operation. I see two problems with this statement. The first is that it ignores that the number of required qubits is larger than in parametrized quantum circuits without a layer structure. The second is that the two-qubit operation between two layers entangles both qubits and errors on one qubit can affect the other as well.

3) In the abstract, the authors mention advantages of the network construction they explore. It should be clarified, in comparison to which alternative approach the advantages are claimed. The question whether there is a quantum advantage, i.e. an advantage over classical machine learning, is to my knowledge open, at least for these constructions. Whether there is an advantage compared to other quantum neural network constructions, is an interesting question. This is partly addressed in <https://arxiv.org/abs/2208.06198>. Here however there is also an application to a quantum chemistry problem. It would be important to explain whether this is a useful application, i.e. whether better performance than with other parametrized quantum circuits can be expected. How do the present results compare to those of ref 41, which is already 5 years old.

4) Whereas it is clear that the approximation of a quantum channel as in the first example is supervised learning, I do not fully understand what is done in the quantum chemistry example. The authors need to explicitly say what the cost function is in both cases. Is this the energy expectation value in the 2nd case? Then this would more be like reinforcement learning.

5) Since the authors call the gates perceptrons, it might be worth mentioning that different types of quantum perceptrons have recently been shown by various groups. To my knowledge, these include <https://arxiv.org/abs/2212.10742>, <https://journals.aps.org/prresearch/pdf/10.1103/PhysRevResearch.4.033190> and <https://arxiv.org/abs/2111.08977>.

Reviewer #2 (Remarks to the Author):

The manuscript presents an impressive experimental demonstration of the operation of quantum

neural networks supplemented with backpropagation. The authors showed that their quantum neural network can learn to predict ground states of molecular Hydrogen as well as to learn quantum channels.

My most important concern is about the scalability of the procedure due to the classical representation of the states during backpropagation and the need for a tomographic reconstruction of the state at each backpropagation step. It is important that the authors explain the challenges of performing backpropagation in the main text and why it is performed classically. The scalability of the current methodology should also be discussed in the main text, as well as the future scalability of the procedure once backpropagation is possible to do experimentally.

I suggest that an expanded version of the sentence in the supplementary material should be included in the main text to emphasize the importance of this limitation: "In this paper, we carry out the backward channel on a classical computer due to the experimental challenges in preparing the quantum states for the backward terms σ . We expect an efficient proposal for the experimental implementation of the backward process, which is important and remains as a future work."

In the chemistry example:

I am confused about this example. The ground state of molecular hydrogen is a pure state. Generally the quantum neural network will produce mixed states on the output layer. If I understand correctly the final state should be disentangled from the rest of the system which means that the extra layers are not necessarily useful. What would seem critical for the description of the system is entanglement between qubits within a layer which is not directly modelled as there are no intra layer gates. Can the authors clarify this point which is confusing me? Since the authors perform tomography, can they check if the output state is disentangled from the rest of the qubits?

Finally, the term "epoch" should be replaced with "iterations" since there are no data sets, and epoch specifically refers to a concept that depends on the size of the training data.

Overall, the experimental results are impressive, but the use of tomography and classical computers for backpropagation represents a limitation of the study. If the authors provide a detailed and convincing explanation as to why these barriers will be overcome in the future, I would be happy to recommend the paper for publication.

Reviewer #3 (Remarks to the Author):

Pan et al. report on experiments performed on a 6-qubit superconducting quantum processor, in which they train a parametrized quantum circuit realized as a quantum neural network (QNN). They study their parametrization and training scheme on three different examples: Learning a two-qubit target channel, finding the approximate ground state energy of a two-qubit Hamiltonian, and learning a single-qubit target channel. They also study the influence of device imperfections such as residual ZZ coupling and qubit decoherence on the performance of the QNN training.

Scientific achievements

The paper provides a first experimental realization of the scheme proposed in Ref.23. By studying multiple different training examples for the QNN algorithm, the manuscript provides practical evidence for possible use cases of the QNN scheme and studies the performance under current NISQ conditions. The authors also derive a general property of this particular QNN ansatz, showing that training can be done on each layer separately without relying on quantum coherence persisting throughout the entire process. While this relaxes some of the requirements in terms of device performance it raises the question how powerful the scheme can generally be. It also puts a question

mark behind the term “deep quantum neural network”. While the network itself is “deep”, quantum coherence during individual runs is relevant only in shallow circuits.

Quality of technical realization

Tuning up all the required gates on a device with enhanced connectivity (sets of four qubits are all-to-all-connected via common bus resonators) is a complex task. However, in order to judge the device performance more quantitatively, I would be interested in seeing standard benchmarking figures (RB, XEB, or process tomography) of the individual building blocks (1Q gates, 2Q gates). The results of the trained processes in the low 90% range seems not particularly impressive given that it is the result of a variational optimization procedure. Without being stated explicitly in the main text, the numerical simulations discussed in Note 2 of the SM seem to predict much higher fidelities on the 99% level. On the other hand, the authors don't state clearly which model they have used for their numerical simulations. Have decoherence and residual ZZ coupling been included? The paper is lacking a rigorous and quantitative discussion about the experimentally obtained fidelities and the main sources of infidelities.

Quality of the presentation

Overall the main manuscript is easily accessible. The introduction provides a good overview over previous work and provides clear motivation for the work. However, important information is missing in the main text, some of which is hidden in the supplementary material.:

1) The backpropagation is carried out on a classical computer, which gets mentioned in the main text only as a side note. However, this is an important limitation of the current experiment. In the present form the experiment could not provide any quantum advantage because the cost for running the process in backwards direction on a PC is equally costly as running it in forward direction. Unless there was a possibility to run the network in backwards direction on the quantum hardware there could be no advantage by performing the forward process on quantum hardware. I would expect a statement such as

“In this paper, we carry out the backward channel on a classical computer due to the experimental challenges in preparing the quantum states for the backward terms. We expect an efficient proposal for the experimental implementation of the backward process, which is important and remains as a future work.”

To be mentioned in the main text instead of the supplemental material.

2) Which target channels are chosen for the training is not specified in the main text. The supplemental document explains that the target channels are constructed using the exact same ansatz as the parametrized QNN ansatz. A more independent choice of target channels could have avoided potential biases. The main text would benefit from a discussion about the construction of the target channels.

3) The data in Fig.3b is lacking a proper discussion in the main text about the control parameters which have been varied in the experiment both to change the effective ZZ coupling strength and the effective qubit lifetime. I was also surprised about the statement that qubit coherence plays a small role. This would imply that most of the imperfection is due to control errors? This relates to my previous comment about the lack of a quantitative discussion about sources of errors.

Conclusion

In the present form I cannot recommend the manuscript for publication. Provided substantial improvements of (i) the presentation and clarity of the manuscript and (ii) the quantitative analysis of experimental performance I would recommend reconsideration for publication in Nat. Comm.

List of major changes (marked in red in the main text and the Supplementary Information):

1. In the main text, we have added some discussions about the experimental challenge for realizing backward channels and the scalability of the quantum backpropagation (BP) algorithm.
2. In Methods of the main text, we have added a detailed description to explain how the control parameters are varied in our simulation to change the effective ZZ interaction strength and the effective qubit lifetime.
3. In the main text, we have added a quantitative discussion about numerical results in Fig.3b and about main sources of experimental imperfections.
4. In the main text, we have added a description of the construction of the target quantum channel.
5. In the main text and the Supplementary Information, we have replaced the term “epoch” with “iterations” in all figures.
6. In Supplementary Note 4, we have added the experimentally characterized two-qubit gate fidelities in TABLE. S2.
7. In Supplementary Note 1, we have added detailed discussions about the experimental proposal to implement backward channels.
8. In the Supplementary Information, we have added a new section (Supplementary Note 2) to discuss about the efficiency of the quantum BP algorithm in training DQNNs in terms of the required number of copies of each training data per training round.
9. We have added other necessary revisions throughout the whole manuscript to improve the presentation and address the three Referees’ comments/suggestions/questions.

Response to Referee #1:

We sincerely thank the Referee for their time on reviewing this manuscript. The Referee has provided a carefully-written report, including a summary of the main advances in our experiments, a discussion about the statements in the manuscript, and their concern about the capability of the structure of DQNNs with current hardware. We took the Referee’s comments and suggestions seriously. Based on their report, we clarified several important points and improved the presentation of this manuscript. Our detailed point-by-point response to the referee’s comments/suggestions is provided in the following.

Comment 1 of Referee #1: “The manuscript discusses the implementation and training of quantum neural networks, in which two-qubit gates are implemented between the qubits belonging to different layers. This type of neural network has been introduced in Beer et al Nat Commun. 11, 808 (2020). In fact studies of the trainability and generalization capacities of this type of quantum neural network, <https://arxiv.org/abs/2208.06198>, found that it performs better than other constructions in many aspects.

The present work could thus form a significant step towards the development of quantum neural networks.

There are however a number of aspects that need to be clarified before I dare to make a recommendation. ”

Authors’ response: We thank the Referee for their accurate summary of the main results of our work. We also thank the Referee for judging that “The present work could thus form a significant step towards the development of quantum neural networks”.

As the Referee noted, it has been found in [<https://arxiv.org/abs/2208.06198>] that the quantum neural network with the layer-by-layer structure can perform better than other constructions in many aspects. We agree with this and have cited this paper in our revised manuscript.

Comment 2 of Referee #1: “1). The work shows supervised learning for the network to train it to approximate a quantum channel. I however did not find what this quantum channel is. To assess the quality of this achievement it is however important to know act kind of channel this is. In particular, the fidelity of the approximation without training, i.e. for random parameters, is already above 70%. Is the targeted output state a pure state? In that case I would expect the untrained fidelity to be much lower, 25%. Also in the deeper network, the untrained fidelity is above 50%.”

Authors’ response: We thank the Referee for raising this important point. In the subsection “Training procedures” of Supplementary Note 1, we have a description about the construction of the target quantum channels. The target channels are constructed using the same ansatz as the DQNN ansatz in our work. For example, we experimentally implement a three-layer DQNN with two qubits in each layer, which is denoted by $DQNN_1$. $DQNN_1$ is used to learn a two-qubit target channel. The target quantum channel is constructed using the same ansatz as $DQNN_1$ with randomly chosen parameters θ_t .

For learning a random 2-qubit target quantum channel, the mean fidelity in the initial training iteration are expected to be around 25%. However, according to experimental results for learning a two-qubit quantum channel shown in Fig.2 in the main text, the initial mean fidelity is above 60% for 30 different initial parameters (only 10 of them are plotted in the figure for clarity), which is relatively high. Additionally, from numerical results shown in Supplementary Fig. S1, we observe that the mean fidelity in the initial training iteration are above 60% for most random parameters. This is due to the fact that the construction of the target channel utilizes the same ansatz as the DQNN ansatz in our work.

As shown in [Nat. Commun. 11, 808 (2020)], DQNNs with the most general form of quantum perceptrons, which consist of arbitrary unitaries acting on all qubits at adjacent layers, have the capability to execute universal quantum computation. However, due to experimental noises, we usually employ restricted forms of perceptrons for implementing DQNNs on noisy quantum devices. For example, in our work, the circuit structure of a quantum perceptron involves two single-qubit rotation gates with variational parameters followed by a fixed two-qubit controlled-Phase gate. Nevertheless, the restricted form of perceptrons could lead to a reduction in the representation power of DQNNs. So in this case, we construct the target quantum channel using the same ansatz as the DQNN ansatz (such a target quantum channel will be within the representation range of the DQNN) for experimental convenience. We remark that the quantum backpropagation (BP) algorithm can also be employed for DQNNs with general forms of quantum perceptrons for learning more generic target channels. The experimental challenge for realizing general forms of quantum perceptrons is to implement generic multi-qubit quantum gates with high fidelity.

In the revised main text, we have added a couple of sentences to discuss the construction of the target quantum channels.

Comment 3 of Referee #1: “(2). The authors argue that this structure enhances the capabilities of current hardware since the coherence of the qubits only needs to be maintained as long as the qubits are involved in a two layer operation. I see two problems with this statement. The first is that it ignores that the number of required qubits is larger than in parameterized quantum circuits without a layer structure. The second is that the two-qubit operation between two layers entangles both qubits and errors on one qubit can affect the other as well.”

Authors’ response: We thank the Referee for raising this important point, which is very helpful for us to improve the presentation. For traditional quantum circuits without the layer-by-layer structure, the coherence time required for qubits scales with the circuit depth. However, for the DQNN with the layer-by-layer structure, qubits in each layer only need to keep their coherence for no more than the duration of two-layer operations, as we can discard the information of qubits in the previous layer after finishing two-layer operations. So in our work, we claim that the layer-by-layer structure enhances the capability of current hardware.

We agree with the Referee that “the number of required qubits is larger than in parameterized quantum circuits without a layer structure”. By reusing qubits, we need at least two layers of qubits to implement the DQNN, thus naturally doubling the required number of qubits for traditional quantum circuits without the layer-by-layer structure. The qubits can be reused as follows: after completing the operations between adjacent layers, we can reset qubits in the previous layer to the fiducial product state $|0 \cdots 0\rangle$, and then reuse them as qubits in the subsequent layer.

The Referee also pointed out that “the two-qubit operation between two layers entangles both qubits and errors on one qubit can affect the other as well”. We agree with the Referee on this point, during two-layer operations, the errors on qubits in the previous layer will affect qubits in the current layer, so the depth of the network will be limited in the experimental implementation due to the experimental noise.

In the revised manuscript, we have clarified these two points in the main text.

Comment 4 of Referee #1: “(3). In the abstract, the authors mention advantages of the network construction they explore. It should be clarified, in comparison to which alternative approach the advantages are claimed. The question whether there is a quantum advantage, i.e. an advantage over classical machine learning, is to my knowledge open, at least for these construction. Whether there is an advantage compared to other quantum neural network constructions, is an interesting question. This is partly addressed in <https://arxiv.org/abs/2208.06198>. Here however there is also an application to a quantum chemistry problem. It would be important to explain whether this is a useful application, i.e. whether better performance than with other parameterized quantum circuits can be expected. How do the present results compare to those of Ref. [41], which is already 5 years old.”

Authors’ response: We thank the Referee for the insightful comments and helpful suggestions. As the Referee pointed out, it is an open question whether there is a quantum advantage over classical machine learning. We fully agree with the Referee on this point. The “advantage” in our abstract refers to the imple-

mentation of the quantum backpropagation algorithm and the less stringent coherence-time requirement for their constituting physical qubits in experiments, rather than the quantum advantage over the state-of-the-art classical counterpart. To avoid possible confusions, we changed the word “advantage” to “merit” throughout the whole manuscript.

In our work, another application is about quantum chemistry problems. As pointed out in [Nat. Commun. 11, 808 (2020)], DQNNs with the most general form of quantum perceptrons, which involve generic unitaries acting on all qubits at adjacent layers, can carry out universal quantum computation. Here, we applied the DQNN to learn the ground state energy of the molecular hydrogen Hamiltonian, as an example. However, it is an open question whether DQNNs have better performance on quantum chemistry problems compared with other parameterized quantum circuits, which remains to be explored. Our DQNNs can be used as a complementary approach to solving quantum chemistry problems.

In ref. [41], the ground state of the molecular hydrogen Hamiltonian can be expressed as a unitary mapping on a simple product state for two qubits, and the unitary mapping can be exactly obtained according to unitary coupled cluster theory. The unitary mapping can be then decomposed into a two-qubit quantum circuit with only one variational parameter, which can be obtained by scanning within a certain range. The experiment results show that for different bond length, the ground state energy can be learned with an error within 1% (Fig.3 in ref.[41]). In contrast, in our work we don’t assume any prior knowledge about the ground state of the molecular hydrogen Hamiltonian, and use a general three-layer DQNN output state with multiple variational parameters as the ansatz to learn the ground state. Additionally, compared to ref.[41], more residual ZZ interactions, qubit decoherence, and gate errors are introduced in our experiments due to more qubits and larger circuit depth. As a result, our experimental results exhibit larger deviation (about 6.7%). However, our theoretical simulation results show that, for 50 different initial parameters, the average error of the learned energies is within 1.3%, which is comparable to the results in ref.[41].

In the revised main text, we have added a couple of sentences to explain that DQNNs can be used as a complementary approach to solving quantum chemistry problems.

Comment 5 of Referee #1: “4). Whereas it is clear that the approximation of a quantum channel as in the first example is supervised learning, I do not fully understand what is done in the quantum chemistry example. The authors need to explicitly say what the cost function is in both cases. Is this the energy expectation value in the 2nd case? Then this would more be like reinforcement learning.”

Authors’ response: We thank the Referee for this helpful suggestion. In our work, The first example is a supervised learning task which aims to learn a target quantum channel. The training goal is to maximize the mean fidelity between the desired output state produced by the target quantum channel τ_x^{out} and the measured DQNN output ρ_x^{out} averaged over all four input states ρ_x^{in} . The second task is to learn the the ground state energy of a given Hamiltonian H . The training goal is to minimize the energy estimate $\text{Tr}(\rho^{\text{out}}H)$ for the output state of the DQNN. As the Referee pointed out, this task can be regarded as a reinforcement learning task.

In the revised main text, we followed the Referee’s suggestion and have added the training goals for both cases.

Comment 6 of Referee #1: “5). Since the authors call the gates perceptrons, it might be with mentioning that different types of quantum perceptrons have recently been shown by various groups. To my knowledge, these include <https://arxiv.org/abs/2212.10742>, <https://journals.aps.org/prresearch/pdf/10.1103/PhysRevResearch.4.033190>, and <https://arxiv.org/abs/2111.08977>.”

Authors’ response: We thank the Referee for bringing these recent interesting papers to our attention. In the revised manuscript, we have mentioned the recent developments of different types of quantum perceptrons, and added the relevant citations.

In summary, we greatly appreciate the Referee’s valuable comments/suggestions, which are very helpful for us to improve the manuscript. Following these comments and suggestions, we have added the description of the construction of the target quantum channel in the main text. In addition, we changed the word “advantages” to “merit” throughout the whole manuscript to avoid possible confusions. We have also carefully addressed all other points raised by the Referee. We hope this substantially improved manuscript will satisfy the Referee and convince them to recommend publication of this work in Nature Communications.

Response to Referee #2:

We thank the Referee for their time reviewing the manuscript and for judging our work “an impressive experimental demonstration of the operation of quantum neural networks supplemented with backpropagation”. They suggested we add a discussion about the scalability of our model and include a detailed explanation of the challenges of performing backpropagation, as well as an explanation of why these barriers could be overcome in the future. In the revised manuscript, we followed these valuable suggestions and have added corresponding discussions. The detailed point-by-point response to the Referee’s comments is provided below.

Comment 1 of Referee #2: “The manuscript presents an impressive experimental demonstration of the operation of quantum neural networks supplemented with backpropagation. The authors showed that their quantum neural network can learn to predict ground states of molecular Hydrogen as well as to learn quantum channels.”

Authors’ response: We thank the Referee for the accurate summary of our work, and greatly appreciate the Referee’s judgment that our work is “an impressive experimental demonstration of the operation of quantum neural networks supplemented with backpropagation”.

Comment 2 of Referee #2: “My most important concern is about the scalability of the procedure due to the classical representation of the states during backpropagation and the need for a tomographic reconstruction of the state at each backpropagation step. It is important that the authors explain the challenges of performing backpropagation in the main text and why it is performed classically. The scalability of the current methodology should also be discussed in the main text, as well as the future scalability of the procedure once

backpropagation is possible to do experimentally. I suggest that an expanded version of the sentence in the supplementary material should be included in the main text to emphasize the importance of this limitation: ‘In this paper, we carry out the backward channel on a classical computer due to the experimental challenges in preparing the quantum states for the backward terms σ^{out} . We expect an efficient proposal for the experimental implementation of the backward process, which is important and remains as a future work.’ ”

Authors’ response: We thank the Referee for raising this very important point and valuable suggestion. In the previous manuscript, for the task of learning a target quantum channel, the optimization goal is chosen to be the mean fidelity between output states given by the DQNN (ρ_x^{out}) and the target quantum channel (τ_x^{out}) averaged over N training data:

$$F = \frac{1}{N} \sum_{x=1}^N F_x(\rho_x^{\text{out}}, \tau_x^{\text{out}}) = \frac{1}{N} \sum_{x=1}^N \left[\text{tr} \sqrt{\sqrt{\tau_x^{\text{out}}} \rho_x^{\text{out}} \sqrt{\tau_x^{\text{out}}}} \right],$$

where $\left[\text{tr} \sqrt{\sqrt{\tau_x^{\text{out}}} \rho_x^{\text{out}} \sqrt{\tau_x^{\text{out}}}} \right]$ is the fidelity between two general mixed quantum states. For each training data, the derivative of the mean fidelity F_x with respect to $\theta_{(i,j),k}^l$ can be expressed as:

$$\frac{\partial F_x}{\partial \theta} = \frac{1}{2} \text{tr} \left(\frac{\partial \rho_x^{\text{out}}}{\partial \theta_{(i,j),k}^l} \cdot \sigma_x^{\text{out}} \right),$$

where $\theta_{(i,j),k}^l$ ($k = 1, 2$) denote the variational parameters of the two R_x gates in the quantum perceptron $U_{(i,j)}^l$, and $\sigma_x^{\text{out}} = (\tau_x^{\text{out}})^{1/2} ((\tau_x^{\text{out}})^{1/2} \rho_x^{\text{out}} (\tau_x^{\text{out}})^{1/2})^{-1/2} (\tau_x^{\text{out}})^{1/2}$. The experimental challenge in realizing the backward channel is preparing the general mixed state σ_x^{out} , especially when the network scales up in the width of the output layer.

In our revised manuscript, we solve this problem by choosing a different optimization goal. The new optimization goal is to minimize the cost function, which is defined to be the average distance between ρ_x^{out} and τ_x^{out} :

$$C = \frac{1}{N} \sum_{x=1}^N C_x(\rho_x^{\text{out}}, \tau_x^{\text{out}}) = \frac{1}{N} \sum_{x=1}^N \|\rho_x^{\text{out}} - \tau_x^{\text{out}}\|_F^2 = \frac{1}{N} \sum_{x=1}^N \text{tr}[(\rho_x^{\text{out}} - \tau_x^{\text{out}})^2],$$

where $\|\cdot\|_F$ denotes the Frobenius norm. Then, the derivative of C_x with respect to $\theta_{(i,j),k}^l$ can be expressed as:

$$\frac{\partial C}{\partial \theta_{(i,j),k}^l} = 2 \text{tr} \left(\rho_x^{\text{out}} \frac{\partial \rho_x^{\text{out}}}{\partial \theta_{(i,j),k}^l} \right) - 2 \text{tr} \left(\tau_x^{\text{out}} \frac{\partial \tau_x^{\text{out}}}{\partial \theta_{(i,j),k}^l} \right). \quad (1)$$

In this way, we only need to experimentally prepare $\sigma_x^{\text{out},1} = \rho_x^{\text{out}}$ and $\sigma_x^{\text{out},2} = \tau_x^{\text{out}}$, which can be directly obtained by running the DQNN and the target quantum channel.

In our work, we experimentally perform the forward process on the quantum processor while implementing the backward channel on a classical computer for the following reason. In the quantum backpropagation (BP) algorithm, the backward channel \mathcal{F}^l produces σ^{l-1} according to $\sigma^{l-1} = \mathcal{F}^l(\sigma^l) = \text{tr}_l \left((\mathbb{I}_{l-1} \otimes |0\rangle_l \langle 0|) U^{l\dagger} (\mathbb{I}_{l-1} \otimes \sigma^l) U^l \right)$. We have provided a new potential experimental proposal to realize \mathcal{F}^l in the Supplementary Note 1, but there could be better strategies, which remain for future work. In

the new experimental proposal to implement the backward channel, we need additional ancillary qubits to prepare maximally mixed states, and separately evaluate two terms in Eq. (1). So the required experimental accuracy is more stringent than that for implementing the forward channel.

The scalability of the current methodology for training DQNNs can be discussed from two aspects. First, as we mentioned in the main text, the layer-by-layer structure only needs to store coherent qubits in two adjacent layers. In other words, the number of coherent qubits required to store only scales with the width of the network and doesn't depend on the depth of the network. Second, to evaluate the gradient of the cost function, we need to carry out multiple measurements for quantum state tomography of ρ_{\pm}^l and σ^l (see Eq. (8) in the Supplementary Information). As more measurements mean more copies of each training data are needed, we use the required number of copies of each training data in a training round as a measure of the efficiency of experimental implementation of the training of the DQNN. This is a crucial parameter especially when producing the training data is expensive. In Supplementary Note 2, we have added a careful discussion of the efficiency of the quantum BP algorithm in training DQNNs. According to the discussion, the required number of copies of each training data scales exponentially with the number of qubits in the hidden layer and the output layer. This is because an exponential number of measurements are required for quantum state tomography and the realization of backward channels. With the quantum BP algorithm, training DQNNs can be more efficient in some cases. In fact, the gradient can be estimated via the SWAP test (without tomographic reconstruction of the state), but this is challenging in current experiments since we need to realize controlled-SWAP gate on large numbers of qubits with high fidelity. We have discussed this point in our response to Comment 5 of Referee #2.

In the main text, we followed the Referee's suggestion and have added an explanation about the experimental challenge in performing the backward process. Additionally, we have added a discussion about the scalability of the quantum BP algorithm. In Supplementary Note 1 and 2, we have provided a more in-depth discussion about the experimental proposal to realize backward channels, the efficiency and the scalability of the quantum BP algorithm.

Comment 3 of Referee #2: "In the chemistry example: I am confused about this example. The ground state of molecular hydrogen is a pure state. Generally the quantum neural network will produce mixed states on the output layer. If I understand correctly the final state should be disentangled from the rest of the system which means that the extra layers are not necessarily useful. What would seem critical for the description of the system is entanglement between qubits within a layer which is not directly modelled as there are no intra layer gates. Can the authors clarify this point which is confusing me? Since the authors perform tomography, can they check if the output state is disentangled from the rest of the qubits? "

Authors' response: We thank the Referee for raising this insightful point. As the Referee mentioned, the ground state of the molecular hydrogen is a pure quantum state. We also agree with the Referee the DQNN will generally produce mixed states in the output layer. So in the training process to approximate the ground state (a pure state), the qubits in the output layer will be gradually disentangled from the other qubits.

There are two points that are confusing to the Referee. The first point is that whether the extra layers are necessarily useful as the final state of the output layer should be a pure state. The second point is that the entanglement within the output layer is not directly modelled since there are no intra-layer gates. We conclude that although there are no intra-layer gates in the network, we can still obtain an entangled pure

Figure 1: **An example of producing an entangled state without intra-layer gates.** Each qubit is initialized in the state $|0\rangle$. First, a Hadamard gate acts on qubit A, then a CNOT gate acts on qubits A and C, where A is the control qubit. Finally, a SWAP gate acts on qubits A and D. The produced output state is $(|00\rangle + |11\rangle)/\sqrt{2}$.

state within the output layer which is disentangled from the other qubits. We consider a simple case. Suppose that qubits A and B are in layer 1, and qubits C and qubit D are in layer 2. There are only inter-layer gates and no intra-layer gates. All qubits are initialized in a simple product state $|0\rangle$. To obtain an entangled pure state within layer 2, we first entangle the qubit A and C by inter-layer gates, and then perform the swap gate on the qubit A and D. The final state of qubits in layer 2 is an entangled pure state which is disentangled from layer 1 (A specific example is given in Fig. 1). So the extra layers are useful which can help to entangle qubits in the output layer if there are no intra-layer gates, even if the output layer will finally be disentangled from the rest of the system.

To verify whether the output state of the DQNN will be gradually disentangled from the other qubits, we choose 50 different initial parameters and classically simulate the training process to learn the ground state energy. We plot the purity of the DQNN output state $[\text{tr}((\rho^{\text{out}})^2)]$ as a function of iterations in the training process. When $\text{tr}((\rho^{\text{out}})^2) = 1$, the DQNN output state is a absolutely pure state disentangled from the other qubits. The results are shown in Fig. 2, which show that for most random initial parameters, the DQNN output state will gradually approach a pure state from an initial mixed state. In our experiments, the measured output state in the last training iteration is not a completely pure state due to experimental imperfections.

In the the revised manuscript, we have clarified this point in the Supplementary Note 1.

Comment 4 of Referee #2: “Finally, the term “epoch” should be replaced with “iterations” since there are no data sets, and epoch specifically refers to a concept that depends on the size of the training data. ”

Authors’ response: We thank the Referee for this valuable suggestion. In our revised manuscript, we have replaced the term “epoch” with “iterations”.

Comment 5 of Referee #2: “Overall, the experimental results are impressive, but the use of tomography and classical computers for backpropagation represents a limitation of the study. If the authors provide a

Figure 2: **Numerical results for the purity evolution of the DQNN output state in the training process to learn the ground state energy of the molecular hydrogen.** The purity of the DQNN output state $[\text{tr}((\rho^{\text{out}})^2)]$ is plotted as a function of iterations in the training process for 50 different initial parameters. The distribution of the final purity is displayed in the inset.

detailed and convincing explanation as to why these barriers will be overcome in the future, I would be happy to recommend the paper for publication. ”

Authors’ response: We greatly appreciate the Referee’s judgment that “the experimental results are impressive”. In the revised manuscript, we provided a new potential experimental proposal to realize the backward channel \mathcal{F}^l in Supplementary Note 1. In the new experimental proposal, additional ancillary qubits are required to realize the backward channel \mathcal{F}^l , and the experimental requirements for implementing the backward channel are more stringent than those for implementing the forward channel. In the future, we expect an experimental implementation of the backward channel on a quantum processor with better performance. In this way, the backward process can be performed without classical computers.

We also agree with the Referee that the use of tomography is a limitation when DQNNs scale up. To evaluate

the gradient of the cost function, we need to carry out multiple measurements for state tomography of ρ_{\pm}^l and σ^l for the evaluation of $h_{\pm} = \text{tr}(\rho_{\pm}^l \sigma^l)$, where $\rho_{\pm}^l = \text{tr}_{l-1} \left(U_{\pm}^l (\rho^{l-1} \otimes |0\rangle_l \langle 0|) U_{\pm}^{l\dagger} \right)$. [see Eq. (8) in the Supplementary Information]. In this way, the required number of copies of each training data scales exponentially with the width of the network, as $O(4^n - 1)$ measurements are required to characterize an n -qubit quantum state.

In the future, we believe that the controlled-SWAP gate that acts on large numbers of qubits can be experimentally realized with high fidelity. Using high fidelity controlled-SWAP gates, we can use the SWAP test to estimate $h_{\pm} = \text{tr}(\rho_{\pm}^l \sigma^l)$. For two mixed states ρ_{\pm}^l and σ^l , the probability of passing the SWAP test is $[1 + \text{tr}(\rho_{\pm}^l \sigma^l)]/2$. In this way, we can directly evaluate the gradient without doing state tomography to obtain ρ_{\pm}^l and σ^l .

In the revised manuscript, we have added the above discussion in the Supplementary Information.

In summary, we greatly appreciate the Referee’s positive evaluation of our work and their very helpful comments/suggestions. Following these comments and suggestions, we have added a discussion about the scalability of our model and included a detailed explanation of the challenges of performing backward channels. We have also carefully addressed all other issues raised by the Referee. We hope that this substantially improved manuscript will satisfy the Referee and convince them to recommend acceptance of this work in Nature Communications.

Response to Referee #3:

We thank the Referee for their time reviewing our manuscript and for pointing out our work “provides a first experimental realization of the scheme proposed in Ref.23”. They suggested we provide standard benchmarking figures of the individual building blocks. In the revised manuscript, we have added the characterized two-qubit gate fidelity for the controlled-Phase gate in each quantum perceptron. They suggested we provide a detailed discussion about numerical results in Fig.3b and about sources of errors. They also suggested we explain the experimental challenges for performing the backward process in the main text. In the revised manuscript, we followed these helpful suggestions and have added corresponding discussions to our manuscript. The detailed point-by-point response to the Referee’s comments is provided below.

Comment 1 of Referee #3: “Pan et al. report on experiments performed on a 6-qubit superconducting quantum processor, in which they train a parameterized quantum circuit realized as a quantum neural network (QNN). They study their parameterization and training scheme on three different examples: Learning a two-qubit target channel, finding the approximate ground state energy of a two-qubit Hamiltonian, and learning a single-qubit target channel. They also study the influence of device imperfections such as residual ZZ coupling and qubit decoherence on the performance of the QNN training.”

Authors’ response: We thank the Referee for the accurate summary of our work.

Comment 2 of Referee #3: “Scientific achievements: The paper provides a first experimental realization

of the scheme proposed in Ref.23. By studying multiple different training examples for the QNN algorithm, the manuscript provides practical evidence for possible use cases of the QNN scheme and studies the performance under current NISQ conditions. The authors also derive a general property of this particular QNN ansatz, showing that training can be done on each layer separately without relying on quantum coherence persisting throughout the entire process. While this relaxes some of the requirements in terms of device performance it raises the question how powerful the scheme can generally be. It also puts a question mark behind the term “deep quantum neural network”. While the network itself is “deep”, quantum coherence during individual runs is relevant only in shallow circuits.”

Authors’ response: We thank the Referee for the accurate summary of our work and pointing out that our work provides “a first experimental realization of the scheme proposed in Ref.23”. We also thank the Referee for noting that the general property of this particular QNN ansatz: “Training can be done on each layer separately without relying on quantum coherence persisting throughout the entire process”.

As for the question of how powerful the scheme can generally be, we refer to Ref.23 for in-depth discussions. As pointed in Ref.23, DQNNs with the most general form of quantum perceptrons, which involve generic unitaries acting on all qubits at adjacent layers, can implement any quantum channel on the input qubits, and are thus capable of carrying out universal quantum computation. In realistic cases, we often use restricted forms of perceptrons due to experimental errors. In general, the restricted form of perceptrons may reduce the representation power of DQNNs.

As the Referee pointed out, regardless of the depth of the network, qubits in each layer only need to keep their coherence for no more than the duration of two-layer operations during each individual run. In our work, the word “deep” in the term “deep quantum neural network” is characterized by the presence of multiple hidden layers, similar to the concept in classical neural networks. The structure of classical deep feedforward neural networks consists of interconnected neurons organized into multiple layers that process input information. The information is passed through one or more hidden layers, undergoing mathematical transformations such as linear operations and activation functions. Similarly, in our DQNNs, each layer contains multiple neurons, where each neuron corresponds to one qubit. The propagation of information between layers is carried out by quantum perceptrons, which are parameterized quantum circuits applied to the qubits in adjacent layers.

In the revised main text, we have clarified that the word “deep” refers to multiple hidden layers, rather than the circuit depth.

Comment 3 of Referee #3: “Quality of technical realization: Tuning up all the required gates on a device with enhanced connectivity (sets of four qubits are all-to-all-connected via common bus resonators) is a complex task. However, in order to judge the device performance more quantitatively, I would be interested in seeing standard benchmarking figures (RB, XEB, or process tomography) of the individual building blocks (1Q gates, 2Q gates). The results of the trained processes in the low 90% range seems not particularly impressive given that it is the result of a variational optimization procedure. Without being stated explicitly in the main text, the numerical simulations discussed in Note 2 of the SM seem to predict much higher fidelities on the 99% level. On the other hand, the authors don’t state clearly which model they have used for their numerical simulations. Have decoherence and residual ZZ coupling been included? The paper is lacking a rigorous and quantitative discussion about the experimentally obtained fidelities and the main

sources of infidelities. ”

Authors’ response: We thank the Referee for raising this important point and valuable suggestion. To judge the device performance more quantitatively, we experimentally characterize the two-qubit gate fidelity for the controlled-Phase gate in each perceptron, which are shown in Table. 1. The average two-qubit gate fidelity is around 98.4%. The characterized fidelities of one-qubit gate (R_x) are above 99.5%.

We also thank the Referee for pointing out that the numerical simulations in Supplementary Note 3 are not stated clearly in the main text about which model is used. In the revised main text, we have clarified that the numerical simulations in Supplementary Note 3 are carried out without considering any experimental imperfections, and are used to benchmark the performance of DQNNs.

We also thank the Referee for pointing out that “the paper is lacking a rigorous and quantitative discussion about the experimentally obtained fidelities and the main sources of infidelities”. To address this issue, we have made the corresponding modifications in this revised version. First, we added a detailed description to explain how the control parameters are varied in our simulation to change the effective ZZ interaction strength and the effective qubit lifetime (two sources of experimental imperfections). Second, we quantitatively discussed the numerical results obtained in Fig. 3(b) and found that the main limitation of the training performance is due to residual ZZ interactions. As a result, the converged experimental values for some initial parameters are in the low 90% in Fig. 2 in the main text, as pointed out by the referee.

In the revised manuscript, we followed the Referee’s suggestion and have added the characterized two-qubit gate fidelities in TABLE S2 in Supplementary Note 4.

Comment 4 of Referee #3: “Quality of the presentation: Overall the main manuscript is easily accessible. The introduction provides a good overview over previous work and provides clear motivation for the work. However, important information is missing in the main text, some of which is hidden in the supplementary material. ”

Authors’ response: We thank the Referee for judging that “the main manuscript is easily accessible” and “the introduction provides a good overview over previous work and provides clear motivation for the work”. In the revised manuscript, we have added some important discussions in the main text. Our detailed point-by-point response to the referee’s comments/suggestions is provided in the following.

Comment 5 of Referee #3: “(1). The backpropagation is carried out on a classical computer, which gets mentioned in the main text only as a side note. However, this is an important limitation of the current experiment. In the present form the experiment could not provide any quantum advantage because the cost for running the process in backwards direction on a PC is equally costly as running it in forward direction. Unless there was a possibility to run the network in backwards direction on the quantum hardware there could be no advantage by performing the forward process on quantum hardware. I would expect a statement such as “In this paper, we carry out the backward channel on a classical computer due to the experimental challenges in preparing the quantum states for the backward terms. We expect an efficient proposal for the experimental implementation of the backward process, which is important and remains as a future work.”

DQNN ₁				
Perceptron	Qubits	Time (ns)	Phase	Fidelity
$U_{1,1}^1$	Q_1, Q_3	62	175°	0.986
$U_{2,1}^1$	Q_2, Q_3	52	180°	0.981
$U_{1,2}^1$	Q_1, Q_4	92	180°	0.978
$U_{2,2}^1$	Q_2, Q_4	82	-155°	0.981
$U_{1,2}^2$	Q_3, Q_6	64	176°	0.988
$U_{1,1}^2$	Q_3, Q_5	64	-117°	0.972
$U_{2,2}^2$	Q_4, Q_6	64	-157°	0.990
$U_{2,1}^2$	Q_4, Q_5	60	-165°	0.982

DQNN ₂				
Perceptron	Qubits	Time (ns)	Phase	Fidelity
$U_{1,1}^1$	Q_1, Q_3	62	175°	0.986
$U_{1,1}^2$	Q_2, Q_3	52	180°	0.981
$U_{1,1}^3$	Q_2, Q_4	82	-155°	0.981
$U_{1,1}^4$	Q_4, Q_6	64	-157°	0.990
$U_{1,1}^5$	Q_5, Q_6	60	-130°	0.985

Table 1: The experimentally characterized two-qubit gate fidelity for the controlled-Phase gate in each perceptron. In our experiments, DQNN₁ denotes the three-layer DQNN, and DQNN₂ denotes the six-layer DQNN. When running DQNN₁ (DQNN₂), we apply the quantum perceptrons in the order from the top to the bottom in the first column. $U_{(i,j)}^l$ denotes the unitary of the quantum perceptron which acts on the i -th qubit at layer $l - 1$ and the j -th qubit at layer l . Each perceptron acts on the qubits listed in the second column. We show the operation time and the rotation angle for the controlled-Phase gate in each perceptron. The experimentally characterized two-qubit gate fidelity for the controlled-Phase gate in each perceptron is displayed in the last column.

To be mentioned in the main text instead of the supplemental material.”

Authors’ response: We thank the Referee for raising this very important point and helpful suggestions. In fact, Referee #2 also provided a similar comment about the experimental implementation of backward process in the quantum backpropagation (BP) algorithm. In our previous manuscript, for the task of learning a target quantum channel, we mentioned that the experimental challenge in realizing the backward channel is preparing the quantum states for the backward terms. In our revised manuscript, this problem is solved by choosing a different cost function. We provided a detailed discussion in the response to Comment 2 of Referee #2 and in Supplementary Note 1.

In our revised manuscript, we also have provided a feasible experimental proposal to realize the \mathcal{F}^l , which produces σ^{l-1} according to $\sigma^{l-1} = \mathcal{F}^l(\sigma^l) = \text{tr}_l \left((\mathbb{I}_{l-1} \otimes |0\rangle_l \langle 0|) U^{l\dagger} (\mathbb{I}_{l-1} \otimes \sigma^l) U^l \right)$. The detail of our experimental proposal is given in Supplementary Note 1. In the new experimental proposal, we need to prepare maximally mixed states (requiring additional ancillary qubits), and separately evaluate two terms in Eq. (1) in our response to Comment 2 of Referee #2. So the required experimental accuracy are higher than that for implementing the forward channel. Therefore in our work, we experimentally perform the forward process of the quantum BP algorithm while simulating the backward process classically. We expect an experimental implementation of the backward channel on a quantum processor with better performance in the future.

In Supplementary Note 2, we also have added a detailed discussion about the efficiency of the quantum BP algorithm in terms of the required number of copies of each training data in a training iteration. We find that the training with the quantum BP algorithm could be more efficient than that without the quantum BP algorithm, when the width of the hidden layer is much smaller than the width of the output layer.

In the revised manuscript, we have added an explanation about the experimental challenge in performing the backward process in the main text. In Supplementary Note 1 and Supplementary Note 2, we have provided a possible experimental proposal to realize backward channels, and a more in-depth discussion about the efficiency of the quantum BP algorithm.

Comment 6 of Referee #3: “(2). Which target channels are chosen for the training is not specified in the main text. The supplemental document explains that the target channels are constructed using the exact same ansatz as the parameterized QNN ansatz. A more independent choice of target channels could have avoided potential biases. The main text would benefit from a discussion about the construction of the target channels.”

Authors’ response: We thank the Referee for raising this important point, which needs to be clarified in the main text. As the Referee noted, the target channels are constructed using the same ansatz as the DQNN ansatz in our work.

As pointed out in [Nat. Commun. 11, 808 (2020)], DQNNs with the most general form of quantum perceptrons, which involve generic unitaries acting on all qubits at adjacent layers, can implement any quantum channel on the input qubits, and are thus capable of carrying out universal quantum computation. In realistic cases, we often use restricted forms of perceptrons for the experimental implementation of DQNNs, since it is challenging to implement generic multi-qubit quantum gates with high fidelity. In our work, the circuit structure of a quantum perceptron is composed of two single-qubit rotation gates with variational parame-

ters followed by a fixed two-qubit controlled-Phase gate. However, the restricted form of perceptrons would reduce the representation power of DQNNs. So we use the DQNN ansatz to construct target quantum channels within the representation range of the DQNN. We remark that the quantum BP algorithm can also be employed for DQNNs with general forms of quantum perceptrons for learning more generic target channels.

In this revised manuscript, we followed the referee’s suggestion and have added a discussion about the construction of the target quantum channels in the main text.

Comment 7 of Referee #3: “(3). The data in Fig.3b is lacking a proper discussion in the main text about the control parameters which have been varied in the experiment both to change the effective ZZ coupling strength and the effective qubit lifetime. I was also surprised about the statement that qubit coherence plays a small role. This would imply that most of the imperfection is due to control errors? This relates to my previous comment about the lack of a quantitative discussion about sources of errors. ”

Authors’ response: We thank the Referee for raising this point. The results in Fig.3(b) are obtained by carrying out numerical simulations. Specifically, we consider two sources of experimental imperfections: decoherence of qubits and residual ZZ interactions between qubits. In our simulation, rotation gates are realized by the evolution of a time-dependent Hamiltonian $H_0(t)$. The unwanted residual ZZ interactions between qubits are taken into account by adding an extra term $H_1(t)$ to $H_0(t)$. The qubit decoherence is taken into account by adding the relaxation term and the pure dephasing term as the collapse operators to the evolution process. In Methods of the main text, we have added a detailed discussion about the numerical simulation, which explains how the control parameters are varied to change the effective ZZ coupling strength and the effective qubit lifetime.

The Referee is confused about the statement that the increase of the coherence time around the experimental characteristic value has a minor effect on the training performance. We note that this observation is for the case when the residual ZZ interaction strength $\mu/2\pi \approx 1$ MHz (close to our experimental characterization). In this case, such a large μ dominantly limits the training performance. This is anticipated given the fact that the average characteristic coherence time of our qubits is above $7.5 \mu\text{s}$, while the total running time of the DQNN is only around $1.2 \mu\text{s}$ (which is significantly shorter). We expect a larger effect of the qubit decoherence when the total running time of the DQNN approaches the average characteristic coherence time. In our experiment, the total running time of the DQNN is estimated by summing the operation times of all controlled-Phase gates and rotation gates. From the numerical results in Fig. 3(b) in the main text, we further observe that the reduction of the residual ZZ interaction strength (around the experimental characteristic value) provides a pretty large improvement on the training performance, also indicating that the main limitation of the experimental performance is due to the residual ZZ interactions.

In the revised main text, we followed the referee’s suggestion and have added a quantitative discussion about the numerical results in Fig. 3(b) and about sources of errors.

In summary, we greatly appreciate the Referee’s invaluable comments/suggestions. Following these comments and suggestions, we have added a discussion about the experimental challenges for implementing backward channels. In the revised manuscript, we have added the characterized two-qubit gate fidelity for the controlled-Phase gate in each quantum perceptron. In addition, we added a detailed discussion about

numerical results in Fig.3(b) and a quantitative discussion about sources of errors in the main text. We have also carefully addressed all other issues raised by the Referee. We hope that this substantially improved manuscript will satisfy the Referee and convince them to recommend acceptance of this work in Nature Communications.

REVIEWERS' COMMENTS

Reviewer #1 (Remarks to the Author):

The revision of the manuscript has now clarified several important aspects of the work. In part these clarifications have revealed that the work is not as impressive as one originally might have thought because important information was only provided in the supplement.

This particularly applies to the fact that the quantum channels that are approximated by the quantum neural network are very similar to the network itself. In some sense the network only learns to approximate itself. The presentation has however been improved now in the sense that the construction of the target channels is described in the main text.

The authors have explained that the need for additional qubits can be circumvented by reusing the qubits of one layer so that the number of required qubits is only twice that of a conventional parameterized quantum circuit. Yet the qubit reset that would be needed to implement the reuse of qubits can't be implemented instantaneously and thus this strategy raises additional coherence requirements for the other qubits that need to "wait" for the reset. Together with the agreement of the authors that coherence of qubits in the preceding layers needs to be maintained as well, I believe it became clear, that the claim that this strategy would have less stringent coherence requirements (as made in the abstract) is not sufficiently founded at this point and hence shouldn't be made.

I agree with the concern raised by other referees that the back propagation is classically implemented and needs a construction of a full density matrix from tomography data, which is not scalable. The authors respond to this and present an alternative cost function. In my understanding, this does however still require the full quantum states and hence non-scalable tomography.

In summary, I still think this is good work and the first realization of quantum neural networks of this type. I thus could still support publication but less enthusiastically than initially. I however think that some claims about the implementation of the quantum analog of back propagation and about less stringent coherence requirements of this platform should not be made in their current form.

Finally I would like to express that it is a bit frustrating for a reviewer if crucial information, such as the target channel in the training, is not provided in the main text.

Reviewer #2 (Remarks to the Author):

The authors have addressed the comments I have raised satisfactorily. The main issue I had with the classical simulation of the backward pass has been addressed in the sense that, even though the authors do not solve the scalability problem in their current implementation, they have discussed a potential method that could work in the future. That said, the authors now clearly state the limitations and scalability issues of their work in the main text.

I recommend the paper for publication.

Reviewer #3 (Remarks to the Author):

Dear editors,

All three referees had expressed very similar concerns about the backpropagation algorithm and the lack of clarity of explanations of the choice of quantum channels in their reports to the previous version of the manuscript.

The authors have very thoroughly and convincingly addressed the concerns raised by the referees and have made substantial changes to the manuscript including the presentation of new results and data.

In the present form, I recommend publication of the manuscript.

List of major changes (marked in red in the main text and the Supplementary Information):

1. We have removed “equipped with backpropagation” in the title of the manuscript.
2. In the abstract, we have added the statement that “we experimentally perform the forward process of the backpropagation algorithm and classically simulate the backward process”.
3. In the abstract of the main text, we have removed the claim “the implementation of the quantum analogue of the backpropagation algorithm”.
4. In the abstract of the main text, we have changed the claim “less stringent coherence-time requirement” to “the number of coherent qubits required to maintain does not scale with the depth of the deep quantum neural network”.
5. In the main text, we have clarified that reusing qubits raises additional coherence requirements for qubits in the current layer, since resetting qubits in the preceding layer takes extra time in experiments.
6. In the main text, we have provided a detailed description of the target quantum channels we aimed to learn in experiments.

Response to Referee #1:

We sincerely thank the Referee for his/her time on reviewing this manuscript and for pointing out that our work is “the first realization of quantum neural networks of this type”. We also greatly appreciate the Referee for supporting the publication of our work. The Referee has provided a carefully-written report, including a discussion about some claims in our manuscript. The Referee also suggested we provide the information for target quantum channels in the main text. We took the Referee’s comments and suggestions seriously. Based on his/her report, we have improved the presentation of this manuscript. Our detailed point-by-point response to the Referee’s comments/suggestions is provided in the following.

Comment 1 of Referee #1: “The revision of the manuscript has now clarified several important aspects of the work. In part these clarifications have revealed that the work is not as impressive as one originally might have thought because important information was only provided in the supplement. This particularly applies to the fact that the quantum channels that are approximated by the quantum neural network are very similar to the network itself. In some sense the network only learns to approximate itself. The presentation has however been improved now in the sense that the construction of the target channels is described in the main text.”

Authors’ response: We thank the Referee for raising this important point. As the Referee pointed out, the target channels are constructed using the same ansatz as the deep quantum neural network (DQNN) ansatz in our work. In our response to the Referee in the first round, we mentioned that DQNNs with the most general form of quantum perceptrons are capable of executing universal quantum computation (shown in [Nat. Commun. 11, 808 (2020)]). However, it is challenging to experimentally realize general forms of quantum perceptrons in current experiments, which requires experimental implementation of the generic multi-qubit

quantum gates with high fidelity. In our work, we experimentally implemented a restricted form of perceptrons. In general, the restricted form of perceptrons could reduce the representation power of DQNNs. So we constructed target quantum channels using the DQNN ansatz, so that the target channel is within the representation range of the DQNN. We note that the quantum BP algorithm can also be used to train DQNNs with general forms of quantum perceptrons for learning more general target channels. In the future, we expect an experimental implementation of the DQNNs with the most general form of quantum perceptrons on a quantum processor with a higher fidelity.

Comment 2 of Referee #1: “The authors have explained that the need for additional qubits can be circumvented by reusing the qubits of one layer so that the number of required qubits is only twice that of a conventional parameterized quantum circuit. Yet the qubit reset that would be needed to implement the reuse of qubits can’t be implemented instantaneously and thus this strategy raises additional coherence requirements for the other qubits that need to “wait” for the reset. Together with the agreement of the authors that coherence of qubits in the preceding layers needs to be maintained as well, I believe it became clear, that the claim that this strategy would have less stringent coherence requirements (as made in the abstract) is not sufficiently founded at this point and hence shouldn’t be made. ”

Authors’ response: We thank the Referee for raising this important point. As the Referee pointed out, it takes extra time in experiments to reset qubits to the fiducial product state when reusing them, which will raise additional coherence requirements for qubits in the current layer. We agree with this point and have clarified it in the revised main text. However, qubits in the current layer only need to keep their coherence for no more than the duration of two layerwise operations (with the preceding layer and the subsequent layer) and resetting qubits (if reusing qubits) in the preceding layer, regardless of the depth of the DQNN. To avoid possible confusion, we followed the Referee’s suggestion and changed the original description “less stringent coherence-time requirement” to “the number of coherent qubits required to maintain does not scale with the depth of the DQNN” in the abstract in the main text.

Comment 3 of Referee #1: “I agree with the concern raised by other referees that the back propagation is classically implemented and needs a construction of a full density matrix from tomography data, which is not scalable. The authors respond to this and present an alternative cost function. In my understanding, this does however still require the full quantum states and hence non-scalable tomography.”

Authors’ response: We thank the Referee for raising this important point, which is very helpful for us to improve the presentation. To evaluate the gradient of the cost function, we need to do state tomography of ρ_{\pm}^l and σ^l to evaluate $h_{\pm} = \text{tr}(\rho_{\pm}^l \sigma^l)$ [see Eq. (8) in the Supplementary Information]. We agree with the Referee that the use of tomography is non-scalable when the network scales up. According to the subsection “Training procedure” of Supplementary Note 1, the use of tomography can be circumvented by using the SWAP test to estimate $h_{\pm} = \text{tr}(\rho_{\pm}^l \sigma^l)$. However, it is challenging in current experiments to realize the controlled-SWAP gate on multiple qubits with high fidelity.

Even with the experimental implementation of SWAP test, our method is still non-scalable, since an exponential number of measurements is required to realize backward channels according to our experimental proposal in Supplementary Note 1. We have clarified this point in the section “Discussion” of the main text.

Comment 4 of Referee #1: “In summary, I still think this is good work and the first realization of quantum neural networks of this type. I thus could still support publication but less enthusiastically than initially. I however think that some claims about the implementation of the quantum analog of back propagation and about less stringent coherence requirements of this platform should not be made in their current form.”

Authors’ response: We thank the Referee for judging that our work is “good work” and pointing out that our work is “the first realization of quantum neural networks of this type”. We also greatly appreciate the Referee for supporting the publication of our work. We followed the Referee’s suggestion and removed the claim “the implementation of the quantum analogue of the backpropagation algorithm” in the abstract in the main text. We also changed the claim “less stringent coherence-time requirement” to “the number of coherent qubits required to maintain does not scale with the depth of the deep quantum neural network” in the abstract in the main text.

Comment 5 of Referee #1: “Finally I would like to express that it is a bit frustrating for a reviewer if crucial information, such as the target channel in the training, is not provided in the main text.”

Authors’ response: We thank the Referee for this helpful suggestion. In the first version of the manuscript, we left the description of the target quantum channels in the Supplementary Information, rather than the main text, in order to save space. We feel embarrassed that this treatment caused some frustrations to the Referee. In our work, the target quantum channel is constructed using the same ansatz as the DQNN ansatz with randomly chosen parameters θ_t . In the revised manuscript, we followed the Referee’s suggestion and have provided a necessary description of the target quantum channels in the main text. However, we think the specific values of θ_t are not crucial, so we put them in Supplementary Table 2.

In summary, we greatly appreciate the Referee’s valuable comments/suggestions, which are very helpful for us to improve the manuscript. Following these comments and suggestions, we have removed the claim “the implementation of the quantum analogue of the backpropagation algorithm” in the abstract and changed the claim “less stringent coherence-time requirement” to “the number of coherent qubits required to maintain does not scale with the depth of the deep quantum neural network” in the abstract. In addition, we have provided a detailed description of the target quantum channels in the main text.

Response to Referee #2:

Comment of Referee #2: “The authors have addressed the comments I have raised satisfactorily. The main issue I had with the classical simulation of the backward pass has been addressed in the sense that, even though the authors do not solve the scalability problem in their current implementation, they have discussed a potential method that could work in the future. That said, the authors now clearly state the limitations and scalability issues of their work in the main text. I recommend the paper for publication.”

Authors’ response: We greatly appreciate the Referee’s very positive evaluation of our work and kind

recommendation of our manuscript for publication in Nature Communications. We thank again for his/her valuable suggestions and comments in the first round, which helped us improve our manuscript significantly.

Response to Referee #3:

Comment of Referee #3: “Dear editors, All three referees had expressed very similar concerns about the backpropagation algorithm and the lack of clarity of explanations of the choice of quantum channels in their reports to the previous version of the manuscript. The authors have very thoroughly and convincingly addressed the concerns raised by the referees and have made substantial changes to the manuscript including the presentation of new results and data. In the present form, I recommend publication of the manuscript.”

Authors’ response: We greatly appreciate the Referee’s kind recommendation of our manuscript for publication in Nature Communications. We thank again for his/her insightful and constructive suggestions/comments in the first round, which have helped us significantly improve both the presentation and the theoretical depth of our work.